# Recent Advances in Anticoagulant Treatment of Immune Thrombosis: A Focus on Direct Oral Anticoagulants in Heparin-Induced Thrombocytopenia and Anti-Phospholipid Syndrome

**DOI:** 10.3390/ijms23010093

**Published:** 2021-12-22

**Authors:** Julie Carré, Georges Jourdi, Nicolas Gendron, Dominique Helley, Pascale Gaussem, Luc Darnige

**Affiliations:** 1Hematology Department, CHU de Poitiers, 86021 Poitiers, France; 2Research Center, Montreal Heart Institute, Montreal, QC H1T 1C8, Canada; georgesjourdi@hotmail.co.uk; 3Faculty of Pharmacy, Université de Montréal, Montreal, QC H3T 1J4, Canada; 4Innovative Therapies in Haemostasis, INSERM UMR-S1140, University of Paris, 75006 Paris, France; nicolas.gendron@aphp.fr (N.G.); pascale.gaussem@aphp.fr (P.G.); luc.darnige@aphp.fr (L.D.); 5Biosurgical Research Lab (Carpentier Foundation), AP-HP, 75015 Paris, France; 6Biological Hematology Department, Assistance Publique-Hôpitaux de Paris-Centre (AP-HP.CUP), 75015 Paris, France; dominique.helley@aphp.fr; 7Paris Cardiovascular Research Center, INSERM UMR-S970, 75015 Paris, France

**Keywords:** heparin-induced thrombocytopenia, antiphospholipid syndrome, immune thrombosis, direct oral anticoagulants, vaccine-induced immune thrombotic thrombocytopenia

## Abstract

For more than 10 years, direct oral anticoagulants (DOACs) have been increasingly prescribed for the prevention and treatment of thrombotic events. However, their use in immunothrombotic disorders, namely heparin-induced thrombocytopenia (HIT) and antiphospholipid syndrome (APS), is still under investigation. The prothrombotic state resulting from the autoimmune mechanism, multicellular activation, and platelet count decrease, constitutes similarities between HIT and APS. Moreover, they both share the complexity of the biological diagnosis. Current treatment of HIT firstly relies on parenteral non-heparin therapies, but DOACs have been included in American and French guidelines for a few years, providing the advantage of limiting the need for treatment monitoring. In APS, vitamin K antagonists are conversely the main treatment (+/− anti-platelet agents), and the use of DOACs is either subject to precautionary recommendations or is not recommended in severe APS. While some randomized controlled trials have been conducted regarding the use of DOACs in APS, only retrospective studies have examined HIT. In addition, vaccine-induced immune thrombotic thrombocytopenia (VITT) is now a part of immunothrombotic disorders, and guidelines have been created concerning an anticoagulant strategy in this case. This literature review aims to summarize available data on HIT, APS, and VITT treatments and define the use of DOACs in therapeutic strategies.

## 1. Introduction

Since 2008, direct oral anticoagulants (DOACs) have been increasingly prescribed for the treatment and prevention of thromboembolic events. Their efficacy and safety are well documented for the prevention of thromboembolism events in the case of non-valvular atrial fibrillation and for the treatment and prevention of venous thromboembolism (VTE) and pulmonary embolism (PE) [1], but their efficacy for immunothromboembolic disorders requires further investigation, particularly in antiphospholipid syndrome (APS) and heparin-induced thrombocytopenia (HIT). DOACs have also been recently proposed for the treatment of vaccine-induced immune thrombotic thrombocytopenia (VITT), which rarely appears after administering adenovirus-vector-based vaccines against the severe acute respiratory syndrome CoronaVirus-19 (SARS-CoV-2). DOACs provide advantages, including oral administration, rapid onset of anticoagulant effects, fixed doses for each indication, fewer drug interactions, and no routine laboratory monitoring. They are thus easier to manage than vitamin K antagonists (VKA) and parenteral anticoagulant drugs; however, they display a long half-life and have limited use in patients with renal or hepatic dysfunction or with altered gut absorption. In this paper, we aim to summarize available data regarding the use of DOACs in immunothrombotic diseases with a focus on HIT, APS, and VITT.

## 2. Differences and Similarities between HIT and APS

HIT and APS are two immunothrombotic disorders with challenging anticoagulation strategies. Both are responsible for venous and/or arterial thrombosis affecting large vessels and microcirculation, and APS is sometimes accompanied by obstetrical morbidity, mainly recurrent fetal loss [2,3]. The prevalence of HIT is estimated at 20,000 cases/year in the USA (1/1500 to 1/5000 hospitalizations) [4], while the incidence of APS is estimated at 40–50 cases/100,000 persons [5]. The main characteristics of the two diseases are reported in Table 1.

In APS, the immune reaction is mediated by antiphospholipids (aPL) antibodies directed against cardiolipin, β2-glycoprotein I (β2GPI), and/or phosphatidylserine/prothrombin (PS/PT) complexes. These antibodies have the potential to prolong coagulation time in laboratory tests, which is called lupus anticoagulant (LAC) activity. Cardiolipin is an anionic phospholipid (PL) of the mitochondrial membrane also present in plasma, while β2GPI (also called apolipoprotein H) is a five “sushi domains” glycoprotein synthesized by the liver, with a high affinity for negatively charged molecules such as cardiolipin. The conformation of platelet factor 4 (PF4) is modified when bound to heparin, whereby β2GPI changes its 3D conformation secondary to cardiolipin bonding via its domain 5, from a close (O shape) to an open conformation (J and S shape), leading to the exposure of pathogenic antibodies binding-sites on domain 1 [10]. These antibodies induce a dimerization of the glycoprotein, which further increases β2GPI affinity for negatively charged PL, triggering various membrane receptors (Apolipoprotein E Receptor 2, GPIb, Toll-like receptors, GPVI) and intracellular signaling pathways, thereby resulting in the activation of many kinases [11,12,13] (Figure 1).

Diagnosing these two pathologies requires further expertise than a single detection of immunoglobulins (Ig). Indeed, HIT antibodies must be able to activate platelets as evidenced by platelet functional assays (namely serotonin-release assay (SRA), heparin-induced platelet aggregation (HIPA), or light transmission aggregometry) [15], while in APS, aPL must persist for at least 12 weeks to be considered as a diagnostic criterion [16]. 

The link between Ig emergence and thromboembolic events regards a multicellular activation. In HIT, anti-PF4/heparin (PF4/H) antibodies activate platelets upon binding with FcγRIIA receptors, thereby triggering the secretion of granules and production of membrane microparticles, ultimately resulting in thrombocytopenia (Figure 2). 

In APS, a mild thrombocytopenia (platelet count between 30 and 100 G/L) is observed in 22% of cases [17], which appears secondary to antibodies directed against platelet membrane glycoprotein (GP) IbIX and GPIIbIIIa [18,19]. Pardos-Gea J et al. recently have associated thrombocytopenia with poor long-term prognoses, where thrombocytopenic patients have a higher risk of death secondary to thrombosis (15% vs. 1%) [20]. Moreover, the onset or worsening of thrombocytopenia may be a sign of thrombosis or a catastrophic APS. A recent study also showed that anti-prothrombin (aPT) antibodies inducing LAC activity are able to trigger platelet activation mediated by FcγRII [21]. In APS and HIT, monocytes and endothelial cells are also activated, leading to tissue factor expression and coagulation activation, thereby increasing thrombin generation and the release of procoagulant microparticles (Figure 1 and Figure 2). Moreover, neutrophil adhesion to endothelial cells is enhanced, resulting in neutrophils extracellular traps (NETs) formation (Figure 1). Another molecular aspect in APS pathophysiology is the modulation of microRNA (miRNA) levels, which results in the increase of tissue factor expression in monocytes and in the development of a pro-inflammatory response whereby miRNA effects pro and anti-inflammatory cytokines levels [22].

HIT and APS, therefore, represent procoagulant disorders that may be life-threatening and that require rapid management, including primary or secondary prevention of thrombosis. In HIT, 20% to 50% of patients who develop thrombocytopenia suffer from new or progressive thromboembolic complications [4].

## 3. HIT: Diagnosis and Standard of Care

Diagnosing HIT is of crucial importance and remains challenging. HIT diagnosis is based on clinical and biological features, and the risk of developing HIT is high (>1%) in cases of unfractionated heparin (UFH) treatment in medical, surgical, obstetrical, or circulatory assistance contexts. This risk is intermediate (0.1%–1.0%) with prophylactic UFH in medical or obstetrical contexts, in cases of low molecular weight heparin (LMWH) treatment after surgery, or as thromboprophylaxis in cancer. Finally, the use of LMWH in medical or obstetrical contexts, either at prophylactic or curative doses, only exposes patients to a low risk of HIT (<0.1%) [2]. 

HIT should be suspected whenever the platelet count drops by 50% or when new thrombosis occurs in a patient 5 to 14 days after beginning heparin therapy. The 4Ts score represents a reliable prediction tool that considers the degree of platelet count decrease, the time to onset of platelets decrease, the occurrence of thrombosis while receiving heparin treatment, and the presence of other causes of thrombocytopenia [2] (Table 2). When the 4Ts score is 3 or less, the probability of HIT is low, and heparin treatment can be continued; with a score of 4 or 5, the probability of HIT is intermediate, while a 4T score ≥6 highly favors HIT [23].

When the HIT probability is intermediate to high, an anti-PF4/H antibodies enzyme-linked immunosorbent assay (ELISA) must be performed as soon as possible, which conveys an accurate negative predictive value (varying between 96.5% and 98.9%, depending on the kit used) and can quickly rule out HIT diagnosis [24]. When ELISA is positive, functional platelet assays should be performed in order to confirm the capacity of anti-PF4/H antibodies to activate platelets from healthy donors in the presence of heparin, thereby confirming the HIT diagnosis [2,25]. SRA is a common functional assay and is considered the gold standard for HIT diagnosis. This assay measures serotonin release from platelets previously incubated with radioactive serotonin in the presence of low or high heparin concentration or in the absence of heparin [26]. Other tests are available, including light transmission aggregometry, HIPA [27,28], and flow cytometry-based assays [2,29]. Functional assays confirm HIT diagnoses when platelet activation is observed only in the presence of a low concentration of heparin, thereby indicating that the platelets’ activation by HIT antibodies depends on heparin.

As soon as HIT is clinically suspected, with at least an intermediate probability 4Ts score, heparin treatment must immediately be ceased and replaced by a non-heparin agent [2,25], where the latter should have a rapid onset of anticoagulant effect. Until recently, the only therapeutic options for these patients were parenteral anticoagulant drugs, including argatroban, bivalirudin, danaparoid, and fondaparinux. The choice between these compounds depends on the clinical setting and some patients’ features [2,25]. Anticoagulation with therapeutic doses is recommended by the American Society of Hematology (ASH) in cases of acute isolated HIT, meaning a HIT diagnosed within the last month, during which the platelet count is ≤150 G/L and platelet-activating anti-PF4/H antibodies are most often present with high thrombotic risk [25], or HIT-associated thrombosis (HITT), except in patients with high bleeding risk, for whom a prophylactic dose is recommended [25].

In HIT patients with a critical illness, high bleeding risk, or high potential need for urgent procedures, the intravenous (IV) direct thrombin inhibitors (DTI) bivalirudin (off-label) or argatroban (Food and Drug Administration (FDA)-approved) should be preferred due to their shorter half-lives [25]. They are widely used despite low evidence of effectiveness with contrasting results (reduction of thrombosis-related mortality together with increased bleeding events) [30]. Argatroban is indicated in patients without hepatic dysfunction and is recommended as a first-line treatment by the American College of Chest Physicians guidelines [31]. The British Society for Haematology recommends argatroban or danaparoid as a first-line treatment [32], while no preferred agent is reported in the ASH guidelines [25]. The French Working Group on Perioperative Haemostasis (GIHP) recommends argatroban as a first-line treatment except for patients with severe hepatic impairment.

Danaparoid is a mixture of heparan, dermatan, and chondroitin sulfates which potentiates the anticoagulant activity of antithrombin, thereby inhibiting the activated factor Xa and thrombin to a much lesser extent. Danaparoid has a long half-life (≈25 h) and can be safely used in patients with non-severe renal insufficiency [2] (Table 3). The GIHP does not recommend prophylactic doses of danaparoid for the treatment of acute HIT; instead, curative IV doses should be prescribed, and treatment efficacy should be monitored using specific anti-Xa activity [2].

VKA should not be used in acute HIT, and their introduction must be delayed until normal platelet count recovery. In addition, switching to VKA in an acute HIT setting exposes the patient to a risk of warfarin-induced skin necrosis and gangrene due to the depletion of protein C and protein S [31]. Warkentin and Kelton showed a comparable rate of thrombosis if heparin is immediately switched to VKA or is ceased without any alternative anticoagulant therapy (10/21 vs. 20/36 HIT patients), which suggests that VKA do not represent safe alternative drugs as long as the hypercoagulable state persists [30].

Fondaparinux represents an attractive alternative treatment of HIT and is a subcutaneous factor Xa inhibitor prescribed off-label in HIT patients. It does not require any systematic laboratory monitoring, and the bridging to VKA is simple since fondaparinux does not affect global hemostasis assays, particularly the prothrombin time and the international normalized ratio (INR) values. Fondaparinux is also renally cleared, which means that kidney function should be considered in treated patients.

## 4. HIT: Update on DOACs Use

Except in situations that require the use of short half-life molecules, such as in patients with high bleeding risk or who are likely to undergo surgery, the use of parenteral treatment in HIT presents disadvantages since they require venous access and specialized laboratory monitoring. In addition, the transition from parenteral treatment to oral VKA may prolong hospitalization since VKA initiation requires platelet count recovery. VKA should also begin simultaneously with parenteral anticoagulation, which may complexify the adjustment of INR due to the possible interference of the former with prothrombin time (as in the case of argatroban, for instance).

In this context, the use of DOACs in HIT has recently become a subject of investigation, especially because they provide a rapid onset of action and have a proven absence of cross-reaction with anti-PF4/H antibodies [33].

Numerous case reports, case series, and observational studies describing the use of DOACs in HIT have been published [34,35,36], but there remains a lack of high-quality randomized clinical trials (RCT) evaluating the efficacy and safety of DOACs in HIT patients. In the recently published systematic review and meta-analysis of 92 studies reporting clinical outcomes of patients treated with non-heparin anticoagulants (argatroban, danaparoid, fondaparinux, DOACs, bivalirudin, and other hirudins) for acute HIT, Nilius et al. [37] outlined significant bias in some studies linked to the lack of information about patient treatment, adherence to anticoagulant therapy, the lack of appropriate control groups, and the brief follow-up period. The rate of platelet recovery, risk of new or progressive thromboembolism, or occurrence of major bleeding complications were nonetheless not affected by patient populations (isolated HIT patients vs. HITT patients vs. all HIT and HITT patients), diagnostic testing strategy (PF4/H immunoassays vs. clinical criteria vs. SRA/HIPA), or study design (prospective study vs. RCT vs. retrospective study) which means that this study provides the best level of evidence available, according to the authors. They reported a platelet count recovery (i.e., ≥150 G/L) in 96% (CI95%: 88–99%) of HIT patients treated with DOACs (*n* = 74), where no significant difference with argatroban, danaparoid, or fondaparinux was noticed. Only bivalirudin was associated with a significant decreased rate of platelet count recovery (74%, CI95%: 58–85%) [37]. In 2019, Barlow et al. [38] published an exhaustive compilation of literature data regarding the use of DOACs in HIT, where they regrouped biological and clinical features of 104 patients with probable HIT who were treated with DOACs. The DOAC treatment was initiated before platelet count recovery in half of the cases as a first-line treatment or after initial parenteral anticoagulation. Similar to the findings of Nilius et al., platelet recovery was achieved in 99% of cases within a median time of 7 days (2–60); therefore, DOACs seem to be effective alternatives to parenteral drugs for HIT treatment. Indeed, DOACs prevented new or recurrent thrombosis in 98% of cases, and bleeding complications occurred in only 3% of cases (in patients who presented known risk factors and concomitant anti-platelet drug treatment) in the narrative review of Barlow et al. Nilius et al. similarly reported a rate of thrombosis of 3% (CI95%: 1–8%) and bleeding complications in 1% (CI95%: 0–22%) of DOAC patients (*n* = 124) where no significant difference was found between all anticoagulants studied. Although the aforementioned studies grouped all DOACs, their efficacy and safety may differ according to each drug.

Rivaroxaban was the most studied DOACs in HIT. In 2016, Linkins, LA. et al. [39] evaluated the efficacy and safety of rivaroxaban in HIT patients in a multicenter, single-arm, prospective cohort study, where 5 out of 12 HIT-positive patients received rivaroxaban 15 mg bidaily (bid), 3 of which were thrombocytopenic at treatment initiation. The other 7 patients received a short course of danaparoid or fondaparinux before being switched to rivaroxaban, and 4 out of these 7 patients were still thrombocytopenic when rivaroxaban was initiated. Platelet recovery occurred in a median time of 7 days in 9 of the 10 initially thrombocytopenic patients. The rivaroxaban dosage was reduced to 20 mg daily upon platelet recovery in patients with isolated HIT or after 21 days in those with HITT. One patient had symptomatic recurrent VTE during the 30-day follow-up (extension of apheresis catheter-related thrombosis that may have preceded rivaroxaban introduction). Another patient underwent bilateral lower limb arterial thrombosis, which was not resolved by an anticoagulation switch and increased platelet count. Finally, one gastrointestinal bleeding event occurred in a patient suffering from gastric cancer.

One year later, Warkentin et al. published the results of the Hamilton experience [40] and incremented literature data with a cohort of 16 HIT patients, whose diagnosis was confirmed with SRA and treated with rivaroxaban. Nine patients were still thrombocytopenic when rivaroxaban was initiated (7 received rivaroxaban as a first-line and 2 after a short course of parenteral non-heparin anticoagulation). No thrombotic event occurred in these patients at the end of the 30-day follow-up nor throughout the treatment period (with a median of 3 months and a range from 17 days to more than 1 year). None of the 16 patients required limb amputation, developed major hemorrhages, or died. The authors also analyzed the literature data regarding apixaban and dabigatran usage in HIT patients, which reported 12 apixaban and 10 dabigatran patients who were transitioned to DOACs before platelet count recovery and 1 patient who received dabigatran after platelet count recovery. Only 1 of these 23 patients had a possible thrombotic event while receiving DOACs (multiple strokes, which might have occurred before starting dabigatran). No major bleeding was reported.

Since 2018, different hemostasis societies have agreed on the use of DOACs in stable HIT patients [2,25]. The GIHP suggests that DOACs could be prescribed as a first-line alternative treatment or second-line therapy after a prior administration of danaparoid or argatroban, except in patients with severe renal or hepatic impairment or high bleeding risk. In contrast, in patients who are unstable, have high bleeding risk, or suffer from life- or limb-threatening thromboembolism event, a parenteral anticoagulant with a short half-life (such as argatroban or bivalirudin) should be prescribed in addition to a strict laboratory monitoring of the anticoagulant effect. In acute isolated HIT (i.e., without thrombosis), the ASH recommends rivaroxaban 15 mg bid until platelet count recovery followed by 20 mg once daily given an indication of ongoing anticoagulation. In cases of acute HITT, rivaroxaban is recommended at 15 mg bid for 3 weeks followed by 20 mg once daily for typically 3 to 6 months [25].

## 5. APS: Diagnosis and Standard of Care

Diagnosing APS requires the association of at least one clinical and one biological criterion. According to the Sydney classification [3], clinical criteria for APS are defined as follows: ≥1 episode of venous, arterial, or microvascular thromboembolism in any tissue or organ; ≥1 unexplained fetal loss beyond the 10th week of gestation; ≥1 premature birth before the 34th week of gestation due to eclampsia, pre-eclampsia, or placental insufficiency; or ≥3 unexplained consecutive spontaneous fetal losses before the 10th week of gestation. The biological criteria include the presence of a LAC activity, the detection of IgG or IgM anticardiolipin (aCL) antibody in medium or high titer (i.e., >40 GPL or MPL), or IgG or IgM anti-β2GPI antibody with a titer > 99th percentile. The laboratory criteria should be persistent and remain positive on 2 or more occasions at least 12 weeks apart. No more than 5 years should separate the positive aPL test and the clinical manifestations [3].

Because APS is a thrombotic disorder, anticoagulation plays a key role in its management. The European League Against Rheumatism (EULAR) [16] distinguishes two aPL profiles that differ regarding the risk level of thrombotic and/or obstetric events as well as the modalities of treatment. While isolated aCL or anti-β2GPI antibodies at low-medium titers, particularly if transiently positive, are associated with a low-risk aPL profile, the presence of LAC activity; double (any combination of LAC, aCL antibodies or anti-β2GPI antibodies) or triple (all three subtypes) aPL positivity; or the presence of persistently high aPL titers is associated with a high-risk aPL profile.

A significant proportion of systemic lupus erythematosus and APS patients [41] have auto-anti-PF4 antibodies without a history of heparin treatment. These auto-anti-PF4 antibodies falsely induce positivity of anti-PF4/H ELISA detection, but functional tests are negative, which could represent a problem for managing these patients when investigating the causes of thrombocytopenia. In contrast, aCL can be detected in HIT patients, but their role in thrombotic events in this context must be evaluated with caution [42].

The accidental discovery of an isolated high-risk aPL profile (thus with no history of thrombosis or obstetrical complication) justifies initiating low-dose aspirin therapy as a primary prophylaxis to reduce the risk of a first thrombotic event [16,43]. In APS patients who underwent a first VTE, VKA are the first choice for secondary thromboprophylaxis with a recommended target INR of 2-3 (higher treatment intensity has shown no benefit [44]); however, aPL antibodies can interfere with some prothrombin time reagents, thereby complicating the reliable measurement of INR and consequently the VKA dosage adjustment, and human recombinant thromboplastins should not be used in this case [45]. LAC may also interfere with the INR measured with point-of-care devices, whose use is thus not advised in APS patients with LAC [45]. In cases of arterial thrombosis, VKA remains the first-line therapy with a recommended target INR of 2–3 or 3–4 alone or combined with low-dose aspirin while considering the individual risk of bleeding and recurrent thrombosis. If recurrent venous or arterial thrombosis occurs despite strict adherence to VKA treatment and a well-controlled INR, the addition of low-dose aspirin, increase of INR target to 3–4, or change to LMWH must be considered based on patients’ characteristics [16,46]. Because VKA are contraindicated during pregnancy, the chosen treatment in cases of obstetric APS relies on low-dose aspirin and prophylactic heparin (UFH or LMWH) up to 6 weeks after delivery [16]. Therapeutic heparin dosing should be considered in cases of recurrent pregnancy complications despite the combination of low-dose aspirin and prophylactic heparin or in women with a history of thrombotic APS [16].

## 6. APS: Update on DOACs’ Use

### 6.1. DOACs’ Therapeutic Use in APS

Multiple studies evaluating the efficacy and safety of DOACs for secondary thromboprophylaxis in APS have been published (Table 3), but a high level of incertitude persists regarding their benefit-risk balance in this context. Study populations are often heterogeneous (excluding arterial thrombosis, mixing different biological profiles etc.), as well as the primary endpoints (arterial or venous events, bleedings, all events together).

Like in HIT, most published studies have focused on rivaroxaban with some conflicting results. In RAPS trial [6], patients were treated with rivaroxaban (*n* = 57) or warfarin (*n* = 59), and no thrombotic or bleeding events were observed during the 6-month follow-up. In contrast, TRAPS study [7] was prematurely stopped due to the large number of undesirable events (composite primary endpoint: thromboembolism, major bleeding, and vascular death) in the rivaroxaban group compared to the warfarin group (Hazard Ratio = 6.7 [CI95%: 1.50–30.5]). This trend has also been observed in the 2-year follow-up of TRAPS patients (2 patients out of the 6 who continued DOACs underwent lower limb deep vein thrombosis and stroke) [47]. In this study, the recruited patients were exclusively triple-positive APS patients, which leads to the conclusion that DOACs should not be used in high-risk triple-positive APS patients [48].

In an open-label randomized non-inferiority study performed in 190 thrombotic APS patients, recurrent thrombosis (mainly stroke) occurred in 11.6% rivaroxaban patients versus 6.3% of VKA patients during the 3-year follow-up [8]. A recently published meta-analysis summarized the data of 4 RCT comparing DOAC to VKA for secondary thromboprophylaxis in APS [49], where 23 and 10 thrombotic events were recorded among 282 and 294 patients treated with DOACs and warfarin, respectively. Recurrent thrombotic events and the risk of VTE were not significantly increased between both groups; however, there was an increased risk of recurrent arterial thrombosis with DOACs compared to warfarin. The recurrence of arterial or VTE seems independent of the type of primary events, where no increased risk of bleeding was found between both groups. The meta-analysis of triple-positive patients’ data unexpectedly showed no higher risk associated with the use of DOACs compared to VKA, whereas the risk was substantially higher in the TRAPS study [47].

Results were recently published regarding a multicenter prospective randomized open-label blinded endpoint Astro-APS study comparing apixaban to warfarin in thrombotic APS [9]. Two protocol changes were instituted before the premature end of the study following the enrollment of the 48th patient. Apixaban was initially prescribed at 2.5 mg bid, and after the 25th patient was randomized, the dose was increased to 5 mg bid, and after the 30th patient was randomized, subjects with prior arterial thrombosis were excluded. A higher risk of stroke in the apixaban group was reported in comparison to the warfarin group, but this result should be interpreted with caution due to the low patient accrual and successive protocol modifications. 

Because studies reported conflicting results, international societies’ recommendations regarding the use of DOACs in APS patients might vary, although it is generally agreed not to use DOACs in triple-positive patients. 

The European Society of Cardiology and European Medicines Agency recommends against using DOACs in all patients [50,51], but other societies exhibit more nuanced recommendations. The British Society for Haematology recommends against the use of DOACs for secondary prophylaxis for arterial thrombosis in APS patients or patients with a history of VTE or triple-positive APS, and they suggest not initiating such treatment in non-triple-positive patients. In APS patients with VTE already receiving DOACs treatment, a switch from DOACs to VKA is recommended in those who are triple-positive, while DOACs’ continuation might be considered in non-triple-positive APS patients [52].

Finally, the European Alliance of Associations for Rheumatology agrees that DOACs should not be used as secondary thromboprophylaxis in triple-positive patients or patients with first arterial thrombosis, but it concedes that DOACs could be considered in patients unable to achieve target INR despite strict adherence to VKA or those with contraindications to VKA. Like VKA, DOACs are contraindicated or strongly not recommended during pregnancy or breastfeeding; therefore, they should not be prescribed for patients with obstetric APS [53].

### 6.2. Interferences of DOACs with Lupus Anticoagulant Diagnosis

Since DOACs are increasingly used to treat patients with thrombotic events, and because it is not uncommon to be prescribed prior to aPL diagnosis, accurate aPL testing is mandatory to avoid inappropriately using these anticoagulant compounds in the APS population. The results of detection and quantification of aCL and anti-β_2_GPI antibodies by solid-phase immunoassay are not influenced by the presence of DOACs in plasma or serum samples from patients taking these drugs [54]; conversely, the presence of DOACs, even at very low concentrations, interferes with LAC testing, which is based on the prolongation of PL-dependent clotting times, mainly inducing false-positive results [55]. Due to their heterogeneity, LAC testing should be performed using at least two coagulation assays with differing analytical principles, where the first is based on dilute Russell Viper Venom Time (dRVVT), and the second is derived from activated partial thromboplastin time. Both assays are compromised by the presence of DOAC compounds in tested samples where a more potent effect of rivaroxaban, edoxaban, and dabigatran occurs compared to apixaban on both assays and, more importantly, on dRVVT [56]. Discontinuing DOAC treatment for at least three days for LAC testing may not be safe without bridging with LMWH. To overcome their interference with LAC testing, many options for DOACs’ in vitro neutralization using adsorption products such as activated charcoal (DOAC Stop^TM^, DOAC Remove^®^) or filter devices (such as DOAC Filter^®^) have been proposed along with some diagnostic algorithms [57,58,59]. The LAC/aPL Scientific and Standardization Committee of the International Society on Thrombosis and Haemostasis (ISTH) published updated guidelines highlighting the usefulness of these in vitro drug adsorption products for reliable LAC testing in DOAC patients [60]. Further investigation of the commercially available adsorption products, however, is still needed to prove the complete neutralization of DOACs in tested samples as well as their neutrality regarding LAC testing.

## 7. Up-to-Date Data on Vaccine-Induced Thrombotic Thrombocytopenia (VITT): A New Context of Immune Thrombosis

The development of vaccines against SARS-CoV-2, and specifically non-replicable adenovirus vector-based COVID-19 vaccines, led to the emergence of VITT, which is a rare but severe reaction to the vaccine that causes the extreme activation of platelets and coagulation with a high risk of death [61]. VITT is suspected in the presence of clinical symptoms associated with laboratory criteria, meaning a drop in platelet count and high D-dimer levels, and the diagnosis is confirmed using anti-PF4/H ELISA and platelet functional assays. Due to the variable sensitivity of various commercial kits [62], the negativity of ELISA does not necessarily exclude VITT diagnosis, and anticoagulation should be started in all patients with probable or confirmed VITT [63]. Anticoagulant treatments proposed by the ISTH include oral and parenteral DTI (argatroban, bivalirudin, dabigatran), oral factor Xa inhibitors (rivaroxaban, apixaban), or fondaparinux, at therapeutic dosing, even if thrombosis is not confirmed [63]. Heparin should be avoided due to the difficulty of ruling out the cross-reactivity of antibodies present in VITT with PF4/H complexes, and it should be reserved for cases where no other non-heparin anticoagulant is available. For the same reasons in HIT, VKA are contraindicated in this high prothrombotic state, where anticoagulation should be continued as long as the platelet count is low and D-dimers levels are high, usually for 3 to 6 months. High doses of IV Ig are recommended in addition to anticoagulant treatment, and plasma exchange can be considered in severe cases (e.g., severe thrombocytopenia and thrombosis) [64,65]. Note that VITT appeared recently, and these recommendations are subject to change based on future investigations.

## 8. Conclusions

The benefit-risk balance of DOACs has been established for preventing thrombosis in atrial fibrillation patients or for preventing recurrent ischemic events following deep vein thrombosis and PE. Promising results have also been recently published for DOACs regarding the prevention of cancer-associated thrombosis [65].

DOACs’ use in HIT seems highly promising, showing satisfying effectiveness in platelet count recovery and thrombosis events, which may reduce hospitalization times through an early discharge of patients and therefore reduced cost to healthcare systems. However, available studies mostly use data regarding clinically stable patients and do not include critically ill patients for whom parenteral anticoagulation should be preferred.

The results of DOACs use in APS are variable, although they seem less effective than the standard care for secondary thromboprophylaxis in triple-positive patients or in those with a history of arterial thrombosis. Further well-designed RCT are anticipated, especially in patients without APL triple-positivity and/or a history of arterial thrombosis. An international registry of thrombotic APS patients treated with DOACs (OBSTINATE —ClinicalTrials.gov Identifier: NCT04262492) is currently open. 

In conclusion, the role of DOACs in antithrombotic therapeutic strategies has continuously been grown over recent decades; however, further studies are needed to expand our knowledge on DOACs’ use in immune thrombosis.

## Figures and Tables

**Figure 1 ijms-23-00093-f001:**
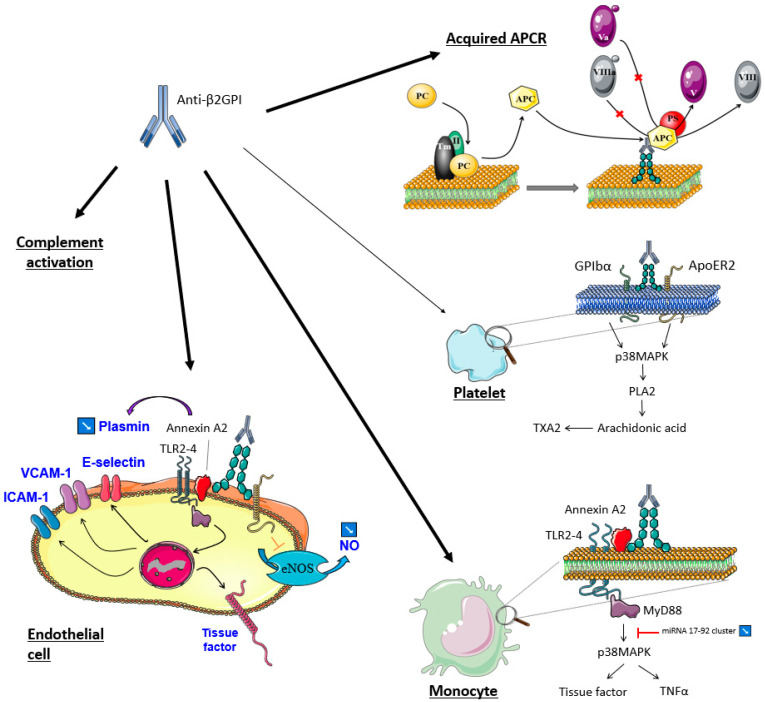
Mechanism of action of anti-β2GPI antibodies in APS. Adapted from Masliah-Planchon J. et al., Rev. Med. Int., 2012 [14]. Anti-β2GPI antibodies bind to negative phospholipids of endothelial cells, monocytes, and, to a lesser extent, platelets. By interacting with membrane receptors (GPIbα, ApoER2, TLR, and annexin A2), they trigger the activation of many intracellular kinases, thereby modulating the expression of procoagulant and anticoagulant molecules. In addition, this binding to membrane phospholipids induces acquired APCR by preventing the formation of coagulation factor complexes on the cell surface which reduces inhibition of factor Va and VIIIa by protein C/protein S complex. These antibodies are also involved in complement activation. APCR: activated protein C resistance; ApoER2: apolipoprotein E receptor 2; GP: glycoprotein; ICAM1: intracellular adhesion molecule; MAPK: mitogen-activated protein kinases; miRNA: microRNA; PC: protein C; PS: protein S; PLA2: phospholipase A2; TLR: toll-like receptor; TM: thrombomodulin; TXA2: thromboxane A2; VCAM: vascular cell adhesion molecule.

**Figure 2 ijms-23-00093-f002:**
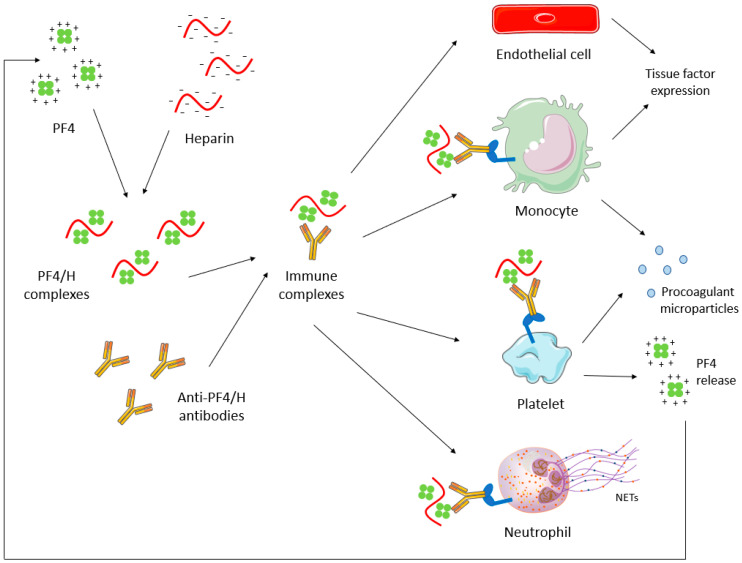
HIT pathophysiology. Antibodies directed against PF4/H complexes induce a multicellular activation, leading to the production of procoagulant microparticles from platelets and monocytes, the tissue factor expression by endothelial cells and monocytes, and the production of NETs by neutrophils. NETs: Neutrophil extracellular traps; PF4: platelet factor 4; PF4/H: PF4/heparin.

**Table 1 ijms-23-00093-t001:** Comparison of HIT and APS.

	HIT	APS
Clinical expression	Thrombosis Thrombocytopenia +++	Thrombosis and/or obstetrical events Thrombocytopenia
Main immunoglobulin isotype	IgG1 and IgG3 (IgM)	IgG2 IgM
Antibody targets	PF4/H complexes	β2GPI-CL PS/PT
Cellular activation	Multicellular activation via FcγRII (on platelets +++, endothelial cells, monocytes, neutrophils)	Multicellular activation via activation cascades of intracellular kinases
	TF expression and secretion of procoagulant microparticles	
Mechanism of platelet activation and thrombocytopenia	Strong platelet activation FcγRII +++	Weak platelet activation F(ab’)2 ++ (FcγRII)
Recommended laboratory tests	Detection of anti-PF4/H antibodies Platelet functional assays	Detection of anti-CL, anti-β2GPI, or LAC activity; twice, 12 weeks apart.
Standard care	Non-heparin treatment (argatroban, danaparoid, bivalirudin, fondaparinux) Contraindication of VKA until platelet count ≥150 G/L	VKA +/− low-dose aspirin LMWH
DOAC use	Recommended in stable patients Rivaroxaban +++	Still debated Contraindicated in triple-positive patients Not recommended in patients with arterial thrombosis
DOAC RCT	None	Cohen H. et al., 2016 [6] Pengo et al., 2018 [7] Ordi-Ros J. et al., 2019 [8] Woller S. et al., 2021 [9]

APS: antiphospholipid syndrome; CL: cardiolipin; β2GPI: β2 glycoprotein I; DOAC: direct oral anticoagulant; HIT: heparin-induced thrombocytopenia; Ig: Immunoglobulin; LAC: Lupus anticoagulant; LMWH: low molecular weight heparin; PF4/H: platelet factor 4/heparin; PS/PT: phosphatidylserine/prothrombin; RCT: randomized controlled trial; TF: tissue factor; VKA: vitamin K antagonist.

**Table 2 ijms-23-00093-t002:** 4Ts pretest clinical score of HIT [23].

	2 Points	1 Point	0 Point
Thrombocytopenia	Platelet count fall > 50% and platelet nadir ≥ 20 G/L	Platelet count fall 30%–50% or platelet nadir 10–19 G/L	Platelet count fall < 30% or platelet nadir < 10 G/L
Timing of platelet count fall	Clear onset between days 5–10; or platelet fall ≤ 1 day with prior heparin exposure within 30 days	Consistent with days 5–10 fall, but not clear; onset after day 10; or fall ≤ 1 day with prior heparin exposure between 30–100 days ago	Platelet count fall < 4 days without recent exposure
Thrombosis or other sequelae	New thrombosis (confirmed); skin necrosis; acute systemic reaction post IV UFH bolus	Progressive or recurrent thrombosis; non-necrotizing (erythematous) skin lesions; suspected thrombosis	None
oTher causes for thrombocytopenia	None apparent	Possible	Definite

IV: intravenous; UFH: unfractionated heparin.

**Table 3 ijms-23-00093-t003:** Proposed molecules as treatment for heparin-induced thrombocytopenia.

	Mechanism of Action	Administration	Half-Life	Clearance
Recommended molecules				
Argatroban	Direct thrombin inhibitor	IV	≈50 min	Hepatic +++
Danaparoid	Factor Xa inhibitor	IV/SC	≈25 h	Renal +++
Bivalirudin	Direct thrombin inhibitor	IV	≈25 min	Renal +++
Fondaparinux	Factor Xa inhibitor	SC	≈17 h	Renal +++
Rivaroxaban	Direct factor Xa inhibitor	PO	5–13 h	2/3 hepatic1/3 renal
Other potential treatments				
Apixaban	Direct factor Xa inhibitor	PO	≈12 h	1/3 renal
Dabigatran	Direct thrombin inhibitor	PO	≈13 h	Renal +++

IV: intravenous; PO: per os; SC: subcutaneous.

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
