# Peer review of "Recent Advances in Anticoagulant Treatment of Immune Thrombosis: A Focus on Direct Oral Anticoagulants in Heparin-Induced Thrombocytopenia and Anti-Phospholipid Syndrome"

_ijms, 2021, doi:10.3390/ijms23010093_

Round 1

Reviewer 1 Report

The authors summarized the available data for HIT and APS in respect to the diagnosis and treatment therapy. In addition, they also mentioned the current data for VITT. The manuscript is well summarized, and the contents are interesting. The information for VITT is also useful for the vaccination to COVID-19. However, there are several points required the revision. The detailed information was shown below.

1. In Table 1, the comparison between HIT and APS was investigated against several parameters. Although this summary table is useful to understand the difference between two disorders, there are some contradiction against the contents in the text.

1) Although the authors clearly described the immunoglobulin isotype, this is just “major” immunoglobulin isotype. Other isotypes are also found. Please add major or some words in the immunoglobulin isotype.

2) As authors showed in the text, the laboratory criteria in antiphospholipid syndrome is lupus anticoagulant, ab2GPI and aCL. Please describe the antibody target based on the guidelines which the authors cited in the text. If the authors would like to add aPS/PT or the related antibodies in antiphospholipid syndrome, please differentiate between aCL/ab2GPI and other related antiphospholipid antibodies.

3) “Biological diagnosis” words are unclear. As the authors showed in the text, HIT is diagnosed based on 4T’s score. Please use words like “Related clinical laboratory tests” or some words. In addition, only lupus anticoagulant, ab2GPI and aCL are recommended for APS diagnosis in guidelines. Please differentiate between the recommendation markers and others.

2. On page 3 line 66, the authors explained the antiphospholipid antibodies including lupus anticoagulant, aCL. ab2GPI and aPS/PT. Please reflect the guideline recommendation.

3. The authors explained serotonin release assay (SRA) on page 4 and showed the assay was the gold standard assay for HIT. However, the detailed information of SRA is not shown in the text. Please add the principle and the diagnosis for HIT if the authors add SRA.

4. Although the authors showed the diagnosis and related markers for VITT, the information is updating. As the limitation. please describe that the contents for VITT are the information when the authors write the manuscript.

5. Please confirm all abbreviation. Although the authors defined "pulmonary embolism" as PE in the introduction section, they use "pulmonary embolism" words again in the conclusion.

Reviewer 2 Report

In my opinion it is necessary to show an impact of miRNA on immune response and several factors which may potentially link miRNA and APS. 
